# Antimicrobial and Antioxidant Properties of Four *Lycopus* Taxa and an Interaction Study of Their Major Compounds

**DOI:** 10.3390/molecules25061422

**Published:** 2020-03-20

**Authors:** Eva Trajčíková, Elena Kurin, Lívia Slobodníková, Marek Straka, Aneta Lichváriková, Svetlana Dokupilová, Iveta Čičová, Milan Nagy, Pavel Mučaji, Silvia Bittner Fialová

**Affiliations:** 1Department of Pharmacognosy and Botany, Faculty of Pharmacy, Comenius University in Bratislava, Odbojárov 10, 832 32 Bratislava, Slovakia; trajcikova1@uniba.sk (E.T.); elena.kurin@uniba.sk (E.K.); nagy@fpharm.uniba.sk (M.N.); mucaji@fpharm.uniba.sk (P.M.); 2Institute of Microbiology, Faculty of Medicine and the University Hospital in Bratislava, Comenius University in Bratislava, Sasinkova 4, 811 08 Bratislava, Slovakia; livia.slobodnikova@fmed.uniba.sk (L.S.); marek.straka@fmed.uniba.sk (M.S.); 3Department of Microbiology and Virology, Faculty of Natural Sciences, Comenius University in Bratislava, Mlynská dolina, Ilkovičova 6, 842 15 Bratislava, Slovakia; 4Department of Molecular Biology, Faculty of Natural Sciences, Comenius University in Bratislava, Mlynská dolina, Ilkovičova 6, 842 15 Bratislava, Slovakia; lichvarikova.aneta@gmail.com; 5Department of Pharmaceutical Analysis and Nuclear Pharmacy, Faculty of Pharmacy, Comenius University in Bratislava, Odbojárov 10, 832 32 Bratislava, Slovakia; dokupilova@fpharm.uniba.sk; 6National Agricultural and Food Centre, Research Institute of Plant Production, 92168 Piešťany, Slovakia; iveta.cicova@nppc.sk

**Keywords:** *Lycopus* leaf infusion, antimicrobial activity, antioxidant activity, interaction study, rosmarinic acid, luteolin-7-*O*-glucuronide

## Abstract

The compositions of leaf infusions of three genotypes of *Lycopus europaeus* L. with origins in central Europe, namely *L. europaeus* A (LeuA), *L. europaeus* B (LeuB), and *L. europaeus* C (LeuC), and one genotype of *L. exaltatus* (Lex), were examined by LC-MS-DAD (Liquid Chromatography Mass Spectrometry and Diode Array Detection) analysis. This revealed the presence of thirteen compounds belonging to the groups of phenolic acids and flavonoids, with a predominance of rosmarinic acid (RA) and luteolin-7-*O*-glucuronide (LGlr). The antimicrobial activity of leaf infusions was tested on the collection strains of Gram-positive and Gram-negative bacteria, and on the clinical *Staphylococcus aureus* strains. We detected higher activity against Gram-positive bacteria, of which the most susceptible strains were those of *Staphylococcus aureus*, including methicillin-resistant and poly-resistant strains. Furthermore, we examined the antioxidant activity of leaf infusions using 2,2-diphenyl-1-picrylhydrazyl (DPPH) and 2,2′-azino-bis(3-ethylbenzothiazoline-6-sulphonic acid) (ABTS) methods, and on NIH/3T3 cell lines using dichlorodihydrofluorescein diacetate (DCFH-DA). We also studied the mutual interactions between selected infusions, namely RA and/or LGlr. In the mixtures of leaf infusion and RA or LGlr, we observed slight synergism and a high dose reduction index in most cases. This leads to the beneficial dose reduction at a given antioxidant effect level in mixtures compared to the doses of the parts used alone. Therefore, our study draws attention to further applications of the *Lycopus* leaves as a valuable alternative source of natural antioxidants and as a promising topical antibacterial agent for medicinal use.

## 1. Introduction

Medicinal plants and herbal preparations have been traditionally used as mixtures of natural substances in phytomedicine for centuries [1]. There exist many good reasons for this practice, such as synergism among constituents, mutual protection of active substances due to their instability, or in some cases, the active substances are still unknown [2]. Synergy or antagonism is present when two or more ingredients of the complex mutually increase or decrease each other’s effect. The final response is higher or lower than a simple summation of single effects (additivity) [3]. This can lead to enhancement in the efficacy of mixtures, reduction of their dosage, and decline of their adverse or toxic effects in clinical practice [4]. The synergistic combinations of agents can be more effective than single-drug treatment thanks to the decrease or even elimination of resistance development, especially in resistant bacteria or viruses [5]. Medicinal plants contain plenty of heterogeneous substances with a positive interaction potential, which makes them a promising area for synergy research.

*Lycopus* L. is a genus of approximately 16 species of flowering plants in the family Lamiaceae, subfamily Nepetoidae [6]. All are herbaceous and native to Europe, East Asia, and North America. The species are often grown in wetlands, damp meadows, stream banks, and other aquatic environments [7]. This genus is characterized by dentate or pinnatifid opposite leaves; flowers in compact, sessile verticillasters in the leaf axils; 4–5 lobed tubulate or campanulate calyces; and a dry, tetrahedral, one-seeded nutlet with corky crests [6,8]. The most important constituents of this genus are essential oil, terpenoids (mostly diterpenes and triterpenes), flavonoids, and phenolic and hydroxycinnamic acids [7,9].

Two species are commonly found in Europe, *Lycopus europaeus* L. and *L. exaltatus* L.f. The first one is a perennial plant native to Europe and Asia and naturalized in the U.S. The plant is odorless and has a bitter taste. Its herbal juice yields a black dye, supposedly used by gypsies to tan their skin, hence the common name of gypsywort. It is also known as the bugleweed, Wolfstrappkraut, bitter bugle, or water horehound [10,11,12].

The biological effects of this plant are attributed to phenolic compounds, mainly to derivatives of hydroxycinnamic acid and flavonoids. According to Fialová et al. [13], the major compounds in leaf infusion are rosmarinic acid and luteolin-7-*O*-glucuronide. Other extracts of the aerial parts revealed a high amount of isopimarane-type diterpenoids and alicyclic diterpenes [9,14,15]. *L. europaeus* is traditionally suggested for treating mild hyperthyroidism with vegetative nervous disturbances. It shows a clear reduction of hyperthyroid symptoms, particularly of cardiac symptoms, and increased body temperature. Extracts of the plant are traditionally also used in tenseness and pain of the mammary gland [15,16,17,18,19]. The methanol extract of the plant showed central and peripheral analgesic and anti-inflammatory effects, which could be associated with the presence of terpenoids, flavonoids, and tannins [11]. The same extract demonstrated significant antitussive activity by inhibition of cough reflex [10]. *L. europaeus* also possessed antiparasitic activity against *T. gallinae* [20], and antioxidant and antimicrobial activity against various bacterial species [12,13,14,15].

*L. exaltatus* is also native to Europe and Asia. Unlike the *L. europaeus*, the leaves of which are only coarsely serrate or sinuate, the leaves of *L. exaltatus* are deeply pinnatifid [21]. The phenolic fingerprints of *L. exaltatus* and *L. europaeus* are similar, as proven by Bucar and Kartnig [22].

Considering the antioxidant, anti-inflammatory, analgesic, and antimicrobial properties of *L. europaeus,* its extracts or isolated compounds might be helpful in the future as potential remedies for the treatment of infections in human or veterinary medicine. Therefore, the aim of our study was a detailed evaluation of the antimicrobial and antioxidant activity of three *L. europaeus* genotypes and one *L. exaltatus* genotype with origins in Central Europe, and to provide phytochemical analysis and fingerprinting of polar phenolic compounds in the leaf infusions. To the best of our knowledge, this is the first study on the biological activities of *L. exaltatus*.

## 2. Results

### 2.1. Identification and Quantification

Secondary metabolites of four *Lycopus* leaf infusions were identified using Liquid Chromatography-Tandem Mass Spectrometry (LC-MS/MS). Acquired MS spectra were compared with authentic standards or with database searches. The maximum allowed mass deviation was 10 ppm. Eleven compounds, including phenolic acids and flavonoids, were found in almost all samples. Identified secondary metabolites are presented in Table 1, along with their retention times (T*_R_*), observed mass in negative ionization mode, MS/MS fragment ions of each secondary metabolite, as well as their quantity in each infusion. The compounds detected in the analyzed infusions were characterized using MS data, together with the interpretation of the observed MS/MS spectra in comparison with standards and data listed in the available literature [13,22].

Peak 1, which presented a pseudomolecular [M − H]^−^ ion at *m*/*z* 191, with all typical product ions presented at *m*/*z* 111, 87, and 85, was identified as citric acid. Danshensu (trihydroxyphenyl propanoic acid) was identified as the second compound with the precursor ion [M − H]^−^ at *m*/*z* 197 and product ions at *m*/*z* 179 and 135. Caffeic acid was identified at a retention time of 21.59 min, with a molecular [M − H]^−^ ion identified at *m*/*z* 179 and product ion at *m*/*z* 135 (which is attributed to the loss of CO_2_), and was compared with an authentic standard. The small peak at 24.58 min is most probably flavonoid quercetin glucuronide with precursor ion [M − H]^−^ at *m*/*z* 477 and with aglycone ion at *m*/*z* 301. Peak 5 showed a precursor ion at *m*/*z* 463 [M − H]^−^ and its MS/MS spectrum presented a product ion at *m*/*z* 301 belonging to quercetin glucoside. Lithospermic acid was detected in 26 min, with a molecular ion [M − H]^−^ at *m*/*z* 537 and typical fragments at *m*/*z* 359, 197, 179, 161, and 135. A small peak (peak 7) presented a pseudomolecular [M − H]^−^ ion at *m*/*z* 521, with fragmentation products at *m*/*z* 359 [M−H−162]^−^, 179, and 161. This peak was identified according to literature as rosmarinic acid hexoside [23]. One of two dominant peaks was recorded in 28.57 min and was identified and compared with authentic standard luteolin-7-*O*-glucuronide (LGlr), with a typical MS [M − H]^−^ ion at *m*/*z* 461 and product ion at *m*/*z* 285. Peak 9, sagerinic acid, a caffeic acid derivative, was identified based on molecular ion [M − H]^−^ at *m*/*z* 719, yielding fragments at *m*/*z* 539, 359, and 161. Peak 10, sulfated rosmarinic acid, was identified according to the literature [23]. It showed the molecular ion [M − H]^−^ at *m*/*z* 439 and a product ion at *m*/*z* 359 due to the loss of 80 Da (sulfated moiety). A small peak at 34.12 min belonged to apigenin-glucuronide, with molecular ion [M − H]^−^ at *m*/*z* 445 and with aglycone ion at *m*/*z* 269. The most dominant peak at 34.48 min revealed the expected presence of the main compound rosmarinic acid (RA), with [M − H]^−^ at *m*/*z* 359. Peak 13 was identified as a caffeic acid derivative, with [M − H]^−^ at *m*/*z* 549 and fragments at *m*/*z* 359 and 161.

The phenolic compounds present in the samples were characterized according to their UV, mass spectra, and retention times, and by comparison with authentic standards when available. For the quantitative analysis, external standards were used. We used LGlr (at λ = 360 nm) for the determination of flavonoids and their derivatives, and RA (at λ = 320 nm) for the determination of phenolic acids and their derivatives. Their calibration curves were obtained by injection of known concentrations (5–100 ppm). Both standards showed good linearity. The following r^2^ values were obtained: LGlr r^2^ = 0.9976, regression curve y = 237.26x − 450.72 and RA r^2^ = 0.999, regression curve y = 340.5x + 297.09. The limit of detection (LOD) of LGlr was 1.89 µg/mL and the limit of quantification (LOQ) was 6.31 µg/mL. The LOD of RA was 1.33 µg/mL and the LOQ was 4.45 µg/mL. The results were expressed in μg per mL of leaf infusion. The examinations of secondary metabolites in *Lycopus* leaf infusions were performed in triplicate. The quantitative results were calculated from calibration curves, expressed as mean values and standard deviation (SD).

### 2.2. Evaluation of Antimicrobial Activity

The antimicrobial activity of the *Lycopus* infusions was tested using microorganisms most frequently isolated from local infections in humans. Collection strains of *Staphylococcus aureus*, *Enterococcus faecalis*, *Pseudomonas aeruginosa*, *Escherichia coli*, *Klebsiella pneumoniae*, and *Proteus mirabilis*, as well as 28 clinical *S. aureus* strains and one collection strain of a yeast *Candida albicans*, were included. For the first screening, 7 well-characterized collection strains were used (for details see the “Material and methods”). The results of antibacterial activities of the four tested *Lycopus* infusions on bacterial collection strains are shown in Table 2.

Staphylococci were the most susceptible; the minimal inhibitory concentrations (MIC) and minimal bactericidal concentrations (MBC) of *Lycopus* infusions were between 2.5 and 5 mg/mL in both methicillin-resistant *S. aureus* (CCM 4750) and methicillin-susceptible *S. aureus* (CCM4223) *S. aureus* strains. The most susceptible Gram-negative bacterial species was *Proteus mirabilis*, with MICs and MBCs between 2.5 and 10 mg/mL. Concerning the yeast strain of *Candida albicans* CCM 90028, it was found to be resistant against any of the samples in the tested concentration range. When comparing the examined plant infusions, the *L. exaltatus* (Lex) infusion was slightly less active than those of *L. europaeus*.

Based on the results with collection strains, the most effective sample—*L. europaeus* B (LeuB)—was chosen for further testing, while clinical strains of *S. aureus* were used as microbial targets. These strains were isolated from infected skin, soft tissue, and wounds (19 strains), and from bloodstream infections (9 strains). Out of 28 strains, 14 were resistant to methicillin or oxacillin (MRSA) and were *mecA* gene-positive. To ensure the appropriate variability of the tested strains, they had various antimicrobial susceptibility patterns and came from different Spa-types. MIC and MBC values of LeuB infusion for clinical *S. aureus* strains are shown in Table 3.

As shown in Table 3, the antibacterial activity of LeuB infusion on clinical *S. aureus* strains was comparable to results obtained with the collection *S. aureus* strains (MICs were found in the interval from 1.25 to 2.5 mg/mL; MBCs reached values from 1.25 to 5 mg/mL). There was no correlation between the MIC or MBC values and the susceptibility of clinical strains to methicillin or to the other antibiotics.

### 2.3. Antioxidant Activity by ABTS (2,2′-azino-bis(3-ethylbenzothiazoline-6-sulphonic acid), DPPH (2,2-diphenyl-1-picrylhydrazyl), and DCFH-DA (dichlorodihydrofluorescein diacetate) Assays

All infusions showed antioxidant activity at the concentration range between 1.25 and 40 µg/mL. The reference compounds, RA and LGlr, possessed antioxidant effects at the concentration range between 0.31 to 20 µg/mL. All tested infusions showed concentration-dependent antioxidant activity. The concentration of a sample necessary to decrease the oxidative stress by 50% (IC_50_) under the experimental conditions was calculated and presented in Table 4 using CompuSyn software. Therefore, a lower IC_50_ value indicates a higher antioxidant activity.

### 2.4. Interaction Analysis by ABTS, DPPH, and DCFH-DA Assays

As we discovered, the most abundant compounds of *Lycopus* leaf infusions were RA and LGlr. Therefore, we choose both substances for interaction analysis. Moreover, we also choose the most effective infusion for each method of antioxidant activity testing (LeuB for ABTS method and Lex for DCFH-DA and DPPH assays, respectively). Equivalent amounts of RA, LGlr, and selected infusions were tested for mutual interactions in mixtures using methods of antioxidant activity testing (ABTS, DPPH, and DCFH-DA). As can be seen in Table 5, all tested combinations showed the synergistic or nearly additive effect. The combination of RA and LGlr showed moderate synergism in all assays. We also provided the interaction analysis between different infusions, which was observed for additive effect only; therefore, we did not present those results here.

DRI values for all tested combinations were higher than 1. This means that the concentration of the samples can be reduced when using the weight equivalent mixtures to achieve the same effect compared to the single-drug treatment. For example, the dose of LGlr can be reduced 5.38 times and RA can be reduced 1.61 times, respectively, when tested in their mixture in the ABTS assay to achieve the same effect as if each compound was used separately. In the latter case, we would need 7.37 µg/mL of LGlr or 2.20 µg/mL of RA; however, when combined together, we need only 1.37 µg/mL each to achieve the effect of a single agent. A significant dose decrease of *Lycopus* leaf infusion is also present in all assays when combined with RA and LGlr. In the ABTS assay, we can reduce the dose of LeuB, RA, and LGlr 7.74 times, 1.95 times, and 6.52 times, respectively. It is more effective to use this triple combination, because to achieve the same effect we needed only 1.13 µg/mL of LeuB, RA, and LGlr; on the other hand, when used alone we need 8.75 µg/mL of LeuB, 2.20 µg/mL of RA, and 7.37 µg/mL of LGlr. Similar dose reduction also occurs in DCFH-DA and DPPH assays. The DCFH-DA assay showed the biggest dose reduction. We needed only 0.19 µg/mL of Lex, RA, and LGlr, as opposed to 2.17 µg/mL of Lex, 0.94 µg/mL of LGlr, and 0.63 µg/mL of RA to achieve the same effect.

## 3. Discussion

Until today, a few studies were published about biological activities of species from the genus *Lycopus*. This was one of the reasons why we have chosen to examine different taxa of *Lycopus* with origins in Central Europe. We analyzed and tested infusions (water extract), the most common form of usage of medicinal plants as remedies in traditional medicine.

Using the LC-MS-DAD (Liquid Chromatography Mass Spectrometry and Diode Array Detection) we revealed the presence of thirteen phenolic compounds and quantified them. There were two compounds with dominant amounts in all infusions: rosmarinic acid (202–237 µg/mL) and luteolin-7-*O*-glucuronide (279–449 µg/mL). Many of the identified compounds were also detected in *L. europaeus* in the past [13]. Next, we looked for the disparities between three genotypes of *L. europaeus* and between different species. We did not detect a high variability in the composition of different genotypes of *Lycopus* between either species, namely *L. europaeus* and *L. exaltatus*, which can also be seen in our results of bioactivities.

The emergence and spread of resistance to antibiotics, as well as the evolution of new strains of disease-causing agents, represent the main factors justifying the need to find and develop new antimicrobial agents. New sources, especially plant sources, are, thus, being investigated. Secondary metabolites form the defense system in plants, which fights against predators such as herbivores, insects, and microorganisms. These substances may act through different mechanisms than conventional antibiotics, and thus they can be utilized in the treatment of infections caused by resistant bacteria. These mechanisms include inhibition of synthesis of nucleic acids, direct interaction with the cell membrane structures of bacteria, which lead to their damage, inhibitory effects on electron transport in the bacterial respiratory chain, and quorum sensing inhibition [25,26]. Several studies on the antimicrobial activity of *L. europaeus* extracts or isolated compounds of several microbial species have been published in the past. They examined both Gram-positive and Gram-negative bacteria as target microorganisms [12,13,14,15]. Gibbons et al. [14] studied an extract and isolated isopimarane diterpenes from *L. europaeus*. None of the tested diterpenes displayed any antibacterial activity, but in combination with tetracycline and erythromycin, a two-fold reduction of the minimal inhibitory concentrations of these antibiotics against two strains of methicillin-resistant *S. aureus*, possessing multidrug resistance efflux pumps, was observed. Radulović et al. [15] isolated a new abietane-type diterpenoid named euroabienol from *L. europaeus* fruits and tested its antimicrobial activity against fifteen strains of bacteria. Euroabienol was most active against *S. aureus* and least active against *E. coli*. On the other hand, *L. europaeus* essential oil, studied by Radulović et al. [12], possessed selectivity towards two Gram-negative strains—*E. coli* and *K. pneumoniae*. Fialová et al. [13] tested the antibacterial activity of *L. europaeus* leaf infusion on *Staphylococcus aureus* clinical strains from catheter-related and skin infections. The infusion showed bactericidal activity at concentrations ranging from 2.5 to 5 mg/mL, without respect to the antimicrobial susceptibility patterns of the tested strains. However, the antimicrobial activity of *L. exaltatus* has not been tested yet. In our study, the tested *Lycopus* infusions were the most active against staphylococci, independently of the antimicrobial susceptibility or the clonal characteristics of the strains. The most susceptible Gram-negative bacterial species was *Proteus mirabilis*. Against the other tested Gram-negative collection strains, as well as the strain of *Enterococcus faecalis* and *Candida albicans*, the examined *Lycopus* infusions were active only at concentrations >10 mg/mL (or were ineffective in the used concentration range). Differences in the susceptibility of various bacterial species were probably based on the well-known differences in their cell walls, such as different permeability due to the different structures of Gram-positive and Gram-negative bacterial cell walls, the charge of the lipopolysaccharide molecules in the Gram-negative cell walls (e.g., *Proteus* spp. versus the majority of the other Gram-negative bacteria), or an insufficient redox potential on the cytoplasmic membrane in *Enterococcus* spp. The reason for the resistance of the tested *Candida albicans* strains might be their similar origin (impermeable cell wall). However, we have not evaluated the interaction of the individual secondary metabolites contained in the *Lycopus* infusions with the cell walls of the mentioned microbes. Concerning the antibacterial activity of the examined *Lycopus* infusions, we suggest that it could be bound to the presence of RA, which even in solo has excellent antimicrobial activity [27,28,29,30]. To equalize the activity of tested infusions and solo RA, we had to triple or even quadruple the doses of infusions [29]. Taking into account the MIC and MBC values of tested *Lycopus* infusions for the bacterial strains used in the study, the infusions might be effective in the treatment of local infections. Tt is known that the application of topical remedies enables an immediate contact of active compounds at rather high concentrations with the microorganisms present in the infectious focus.

Antioxidants have proven beneficial effects on health. There are many studies confirming their ability to prevent diseases, including cancer, neurodegenerative, autoimmune, or cardiovascular diseases [31,32,33,34]. Among different secondary plant metabolites, phenolics and flavonoids are the main compounds with considerable antioxidant activity [35,36,37]. *Lycopus* spp. contains a high amount of phenolics and flavonoids; therefore, it is a perspective genus possessing antioxidant properties. Antioxidant activities of different *Lycopus* spp., including *L. europaeus*, were previously published [13,38,39], but the activity of *L. exaltatus* leaf infusion was explored for the first time. For antioxidant activity evaluation, we used DPPH and ABTS in vitro scavenging assays. Both 2,2-diphenyl-1-picrylhydrazyl (DPPH) and 2,2′-azino-bis(3-ethylbenzothiazoline-6-sulfonic acid) diammonium salt (ABTS) are well-known radicals and radical precursors. They are active in assays based on electron transfer in the protic solvent [40]. We used DCFH-DA on NIH-3T3 cell line for the evaluation of intracellular oxidative stress inhibition by plant infusions. Some external antioxidants could help intracellular antioxidant systems to reduce the damage due to free radicals and reactive oxygen species (ROS). In the chosen method, we used the fluorescence-based indicator DCFH-DA [41,42]. As a generator of oxidative stress, we used H_2_O_2_ and we studied the inhibition of artificially created oxidative stress in the presence of plant samples.

The free radical scavenging activity of *Lycopus* infusions might be related to the content of phenolics, where RA and LGlr were detected as the most abundant ones. Phenolic compounds can easily donate electrons to reactive radicals as DPPH^•^ or ABTS^•+^. RA is a well-known antioxidant and there are many studies that confirm its effects [43,44,45,46]. Luteolin is also known for its antioxidant activity [47]. According to literature, glycosylation of flavonoids reduces their antioxidant activity when compared to the corresponding aglycones [48]. One study compared the antioxidant activity of RA and LGlr using DPPH assay. The antioxidant activity of RA was a few times higher than the effect of LGlr [49], which corresponds to our results. Our results showed that RA is more than three times more effective than LGlr (IC_50_ 2.20 µg/mL and 7.37 µg/mL, respectively) using ABTS assay. LeuB has the highest amount of RA (237.5 ± 0.8 µg/mL) and is also the most effective sample in this method (IC_50_ 8.75 µg/mL). In DCFH-DA and DPPH assays (IC_50_ 0.63 µg/mL and 1.81 µg/mL, respectively), RA is only 1.5 and 1.8 times more effective than LGlr (IC_50_ 0.94 µg/mL and 3.29 µg/mL, respectively). Therefore, we suppose that samples with a noticeably higher amount of LGlr probably possess higher antioxidant activity than other samples. Only one sample (Lex) had a comparable amount of RA as the other samples (216.2 ± 0.1 µg/mL), but 1.6 times more LGlr (449.8 ± 0.4 µg/mL) than LeuB, which was the most effective sample in the ABTS assay. This could be the reason why Lex was the most effective sample in DCFH-DA and DPPH assays (IC_50_ values of 2.17 µg/mL and 8.30 µg/mL, respectively). The second highest amount of LGlr was found in *L. europaeus* (LeuA), which was the second most effective sample in DCFH-DA and DPPH assays (IC_50_ values of 2.64 µg/mL and 8.42 µg/mL, respectively).

Diverse secondary metabolites are present in plants. This heterogeneity assumes a high likelihood of interactions among molecules. Because the mode of action of complex mixtures (including the derivatives from plants) cannot be attributed to a single compound in most cases, interaction analysis, which evaluates the character of the mutual reaction, is necessary. Synergy or antagonism is present if the effect of a sample combination is greater or lesser than from the dose–effect relationships of single agents (i.e., additivity). Because of its complexity, the synergy study has become a key area in phytomedicine research in recent years [40]. For the interaction evaluation, the combination index derived by Chou was used [24]. We applied an interaction analysis between RA and LGlr using ABTS, DPPH, and DCFH-DA assays for the first time. RA and luteolin acted synergistically when testing anti-inflammatory activity in lipopolysaccharide-stimulated RAW 264.7 macrophage cells [50], so there was a chance they might also act synergistically in our models. In our study, the combination of RA and LGlr showed moderate synergism in all assays. In all cases (tested combinations) we detected a synergistic or nearly additive effect, which may support preferable usage of mixtures and combinations for the future.

## 4. Materials and Methods

### 4.1. Plant Material

Aerial parts of the three genotypes of *L. europaeus*, namely *L. europaeus* A (LeuA), *L. europaeus* B (LeuB), and *L. europaeus* C (LeuC), and one genotype of *L. exaltatus* (Lex), were collected in July during sunny weather. The plants were picked up from different places, as shown in Table 6. The harvested plants (about 20–30 individuals) were 1 year old and all of them were collected during their flowering time. The plants were dried at room temperature (25 °C) in shade. The leaves were separated from the stems and flowers.

### 4.2. The Preparation of Infusions

The infusions were prepared according to the article ‘‘Decocta Infusa’’ in the Czech–Slovak Pharmacopoeia, 4th edition [51], as follows: 15 g of leaves were macerated in 150 mL of boiling deionized water for 5 min and then cooled at room temperature for 45 min. The obtained infusion was filtered, frozen, and lyophilized. The yields of infusions were 21.33% for LeuA, 19.77% for LeuB, 17.29% for LeuC, and 20.34% for Lex.

### 4.3. LC-MS-DAD Analyses: Identification and Quantification of the Constituents

The LC-MS-DAD analyses of four samples of *Lycopus* leaf infusions were performed on an Agilent 1260 Infinity LC System (Agilent Technologies, Santa Clara, CA, USA), equipped with a binary pump, an autosampler, a column thermostat, and a diode array detector (DAD), coupled to a quadrupole–time-of-flight (6520 Accurate-Mass QTOF) instrument equipped with an orthogonal electrospray ionisation source (ESI) (Agilent Technologies, Santa Clara, CA, USA). The separation of *Lycopus* infusions was carried out on a Kromasil C18 column (150 mm × 4.6 mm, 5 µm, Sigma-Aldrich, Munich, Germany) at 35 °C and a flow rate of 0.4 mL/min. Deionized water (adjusted to pH 3.1 with HCOOH/NH_4_HCO_3_) and acetonitrile were used as mobile phases A and B, respectively. The following gradient program was used: 5% B (0 min), 20% B (20 min), 20% B (25 min), 40% B (40 min), 50% B (60 min), 65% B (65 min), 100% B (90 min), 100% (95 min), and 5% (96 min). The ESI ion source parameters were as follows: capillary voltage: 3.5 kV; nebulizer: 40 psi (N_2_); dry gas flow: 10 L/min (N_2_); and dry temperature: 300 °C. The mass spectrometer was operated in an auto MS^2^ mode, where each negative ion MS scan (*m*/*z* 100–3000, average of four spectra) was followed by MS^2^ scans (*m*/*z* 100–3000, average of four spectra, isolation window of 4 amu, collision energy 20 eV) of the two most intense precursor ions. Ions were excluded from analyses for 0.5 min after two MS^2^ spectra had been acquired. Nitrogen was used as the collision gas. Phenolic compounds were identified by comparing their UV and mass spectra with literature and authentic standards when available and by measuring accurate *m*/*z* values.

The quantitative determination of phenolic compounds in *Lycopus* leaf infusions was performed by the method of external standards. The HPLC-DAD (High-Performance Liquid Chromatography with a Diode-Array Detector) chromatograms were acquired at two wavelengths of 320 nm and 360 nm, respectively. The standard of RA (purity 96%, Sigma-Aldrich, St. Louis, MO, USA) was used for quantification of phenolic acids and their derivatives (λ = 320 nm) and LGlr (primary RS, purity > 95%, Sigma Aldrich St. Louis, MO, USA) was used as a standard for quantification of flavonoids and their glycosides (λ = 360 nm) (see Table 1). The examinations of secondary metabolites in *Lycopus* leaf infusions were performed in triplicate. The quantitative results were calculated from calibration curves, expressed as mean values and standard deviation (SD).

### 4.4. Antimicrobial Activity Testing

Seven bacterial collection strains (*Staphylococcus aureus CCM 4223, Staphylococcus aureus CCM 4750, Enterococcus faecalis CCM 4224, Pseudomonas aeruginosa CCM 3955, Escherichia coli CCM 3954, Klebsiella pneumoniae* CCM 4415, *Proteus mirabilis* CCM 7188), one collection strain of *Candida albicans* (CCM 90028), and 28 clinical *S. aureus* strains were used in the study. The collections strains were purchased from the Czech Collection of Microorganisms, Brno, Czech Republic (for more detailed characteristics, see Table 7); the clinical strains were isolated from infections of patients hospitalized at the University Hospital in Bratislava, Slovakia. Nineteen strains originated from infections associated with skin, soft tissue, or wounds, and 9 strains came from bloodstream infections (Table 8).

The antimicrobial susceptibility of clinical *S. aureus* strains was detected in oxacillin, erythromycin, clindamycin, tetracycline, cotrimoxazole, ciprofloxacin, and mupirocin, using the disk diffusion method according to the EUCAST recommendations [52]. For the testing, Mueller–Hinton agar (OXOID, Great Britain) and commercial antibiotic disks (OXOID, Great Britain) were used. Resistance to methicillin or oxacillin was confirmed by penicillin binding protein 2a latex agglutination test (MRSA-screen, Denka Seiken Co, Japan) and by PCR (Polymerase Chain Reaction) for *mecA* gene detection [53]. Spa typing was performed according to Ridom Bioinformatics [54]. PCR products were purified by illustra ^TM^ ExoProStar ^TM^ (GE Healthcare, Chicago, IL, United States). The PCR amplicons were sequenced by BigDye Terminator v3.1 Cycle Sequencing Kit (Applied Biosystems, Foster City, CA, USA) and sequencing was carried out on ABI PRISM 3130xl Genetic Analyser (Applied Biosystems, Foster City, CA, USA). DNA sequences were analyzed with BioNumerics software (Applied Maths, Sint-Martens-Latem, Belgium).

The minimal inhibitory (MIC) and minimal bactericidal concentrations (MBC) of plant material infusions were tested by broth microdilution assay according to the EUCAST recommendations [52]. For antimicrobial activity testing, the lyophilized plant infusions were dissolved in sterile Aqua Pro Injection and filtration-sterilized (Merck Millipore Ltd., Ireland). The testing was performed in U-shaped, sterile 96-well microtiter plates (Roll s.a.s., Piove di Sacco, Italy). The samples in the working concentration were added to the starting wells with a double concentrated antibiotic susceptibility testing medium (Mueller-Hinton Broth, OXOID Ltd., Basingstoke, UK). Afterward, serial geometric dilutions from 0.3125 to 5 mg/mL were prepared in the final volume of 100 µL. Bacterial inoculum was prepared from a culture grown overnight on blood agar. Well-isolated colonies were suspended in sterile physiologic solution and the suspension was adjusted to contain 1 × 10^6^ CFU/mL. The standardized suspensions of microorganisms were added to each well in 10 µL aliquots (except for the sterility control wells, i.e., negative controls) and incubated for 24 h at 35 °C. Wells with microorganisms in the medium free of the tested agent were used as growth control (positive controls). MIC was determined as the lowest concentration of antimicrobial agent that completely inhibited the growth of the tested microorganisms, detected by the unaided eye. MBCs were determined by sub-culturing the samples from wells yielding a negative microbial growth. After their overnight incubation at 35 °C on the agar medium free of antimicrobial agents, the MBCs were determined as the lowest concentration at which 99.9% of the tested microbial inoculum was killed.

The activity of the plant infusions on yeast *Candida albicans* was tested by a modified broth microdilution test according to the Clinical and Laboratory Standards Institute document M27-A1 [55]. Then, 100 µL aliquots of yeast cell suspension in physiologic saline, prepared from 24 h culture grown aerobically on Sabouraud agar (BD, Sparks, MD, USA), were added to the samples of 2-fold-concentrated tested plant infusions in 100 µL volume of Sabouraud-broth-modified antibiotic medium 13 (BD, Sparks, MD, USA) in the U-shaped microtiter plate wells. The final concentration of the yeast cells in each well was 1 × 10^4^ yeasts/mL. Microtiter plates were incubated at 35 °C for 48 h aerobically. Samples without yeasts and without tested plant infusions were run in each test plate. The MICs and MBCs were determined using the method described in susceptibility testing of bacterial strains.

All MIC and MBC tests were performed in three independent runs.

### 4.5. Evaluation of Antioxidant Activity

#### 4.5.1. DPPH Radical Scavenging Assay

Total free radical scavenging capacity of the infusions from different plant samples was estimated spectrophotometrically using the stable 2,2-diphenyl-1-picrylhydrazyl radical (DPPH, Sigma-Aldrich, St. Louis, MO, USA), which has an absorption maximum at 515 nm [56]. A solution of the radical was prepared by dissolving 4.4 mg DPPH in 200 mL distilled methanol. Lyophilized plants were dissolved in deionized water in various concentrations, while RA and LGlr were dissolved in distilled ethanol. Here, 25 μL of the samples and 225 μL of DPPH solution were added to 96-well Greiner UV star microplates (Greiner-Bio One GmbH, Frickenhausen, Germany) and mixed. Absorbance was measured after 30 min at the wavelength of 515 nm using an Infinite M200 microplate reader (Tecan AG, Grӧdig, Austria) and compared to a blank. The capability to scavenge the DPPH radical was calculated using Equation (1):(1)$$\mathrm{DPPH}\;\mathrm{scavenging}\;\mathrm{effect}\;\left(\%\right)=\left(\frac{A_{B}-A_{A}}{A_{B}}\right)\times \;100$$
where A_B_ is the absorbance of blank and A_A_ is the absorbance of the sample. The antioxidant activity of plant infusions was tested in quadruplicate and compared with RA, which was used as positive control.

#### 4.5.2. ABTS Radical Scavenging Assay

The free radical scavenging activity of plant samples was determined spectrophotometrically by the ABTS decolorization assay [57]. ABTS radical cations were produced via the reaction of 7 mM water solution of ABTS (purity ≥ 98%, Sigma-Aldrich, St. Louis, MO, USA) with 2.45 mM water solution of potassium persulfate (purity ≥ 99.0, Sigma-Aldrich, St. Louis, MO, USA) (1:1, *V*/*V*), then stored in the dark at room temperature for 16–24 h before use. Then, 1.1 mL of the ABTS radical cation solution was then diluted with distilled ethanol to obtain the final volume of 50 mL. Plant lyophilisates were dissolved in deionized water, while RA and LGlr were dissolved in distilled ethanol in various concentrations. Here, 2.5 μL of samples and 247.5 μL of diluted ABTS radical cation solution were added to 96-well Greiner UV star microplates (Greiner-Bio One GmbH, Frickenhausen, Germany) and mixed. Absorbance was measured after 6 min at a 734 nm wavelength using an Infinite M200 microplate reader (Tecan AG, Grӧdig, Austria) and compared to a blank. Percent inhibition of absorbance at 734 nm was calculated using Equation (2):(2)$$\mathrm{ABTS}\;\mathrm{scavenging}\;\mathrm{effect}\;\left(\%\right)=\left(\frac{A_{B}-A_{A}}{A_{B}}\right)\times \;100$$
where A_B_ is the absorbance of the ABTS radical cation + ethanol, while A_A_ is the absorbance of the ABTS radical cation + sample. The antioxidant activity measurements of plant infusions were performed in quadruplicate and compared with RA, which was used as positive control.

#### 4.5.3. Detection of Intracellular Oxidative Stress

NIH-3T3 cells (mouse embryonic fibroblasts) were obtained from the Department of Pharmacology and Toxicology, Faculty of Pharmacy, Comenius University in Bratislava, Slovakia. The cells were grown in DMEM (Sigma-Aldrich, St. Louis, MO, USA) supplemented with 10% FBS (Sigma-Aldrich, St. Louis, MO, USA), 100 µg/mL streptomycin (Sigma-Aldrich, St. Louis, MO, USA), and 100 IU/mL penicillin (Sigma-Aldrich, St. Louis, MO, USA) at 37 °C in a humidified atmosphere with 5% CO_2_. The cells were sub-cultured twice a week. Detection of oxidative stress with DCFH-DA was determined according to Miranda-Rottmann et al. [58] with slight modifications. The ROS generation was measured using the fluorescence-based indicator, dichlorodihydrofluorescein diacetate (DCFH-DA, purity ≥ 97%, Sigma-Aldrich, St. Louis, MO, USA). The non-ionic DCFH-DA crosses the cell membrane and is hydrolyzed to nonfluorescent DCFH by intracellular esterases. In the presence of ROS, such as hydrogen peroxide, DCFH is rapidly oxidized to fluorescent dichlorofluorescein (DCF), which can be detected using a sensitive fluorescence spectrophotometer. The fluorescence intensity reflects the amount of intracellular ROS formed [59]. The 3T3 cells were seeded in the black 96-well plates (Sarstedt, Nümbrecht, Germany) at 15,000 cells/100 uL/well. After 24 h incubation, the medium was removed and replaced by the serum-free medium (Sigma-Aldrich, St. Louis, MO, USA). After 1 h incubation, 5 µl of the tested infusions (RA and LGlr) in various concentrations were added. The tested samples were dissolved in the serum-free medium. After 1 h, DCFH-DA was added (10 µM in final concentration). After 15 min, H_2_O_2_ (Sigma-Aldrich, St. Louis, MO, USA) was added (100 µM in final concentration). The intracellular fluorescence of DCF was measured by excitation and emission at 480 and 530 nm, and compared to a blank after 15 min in an Infinite M200 microplate reader (Tecan AG, Grӧdig, Austria). All measurements were performed in quadruplicate and compared with RA.

### 4.6. Interaction Analysis of Antioxidant Activities

For quantitative determination of drug interactions, combination index (CI) derived by Chou was used, where CI < 1, = 1, and > 1 indicate synergism, additive effect, and antagonism, respectively [24]. Equation (3) shows the general equation for an n-drug combination at x% inhibition:(3)(CI)xn=∑j=1n (D)j/ (Dx)j
where ^n^(CI)_x_ is the combination index for n drugs that exerts x% inhibition, (D_x_) is a dose “alone” that exerts x% inhibition, and D is a dose “in combination” that inhibits a system by x%.

Using the CI grading system, synergism and antagonism were subdivided by Chou [24] into 11 ranges, as seen in Table 10.

One of the major objectives of using synergistic drug combinations is the reduction of the dose of the drugs needed to achieve the same effect. The dose reduction index (DRI) shows how much the dose of each drug in a synergistic combination can be reduced at a given effect level compared with the doses of each drug alone. The DRI is important in clinical situations, in which dose reduction leads to reduced toxicity, while the effect remains the same [24]. The calculation of DRI is shown in Equation (4):(4)$${\left(DRI\right)}_{1}=\frac{{\left(D_{x}\right)}_{1}}{{\left(D\right)}_{1}};\;{\left(DRI\right)}_{2}=\frac{{\left(D_{x}\right)}_{2}}{{\left(D\right)}_{2}}\ldots etc.$$

The value of DRI > 1 is beneficial because it indicates a dose reduction but does not necessarily always indicate synergism. The greater the DRI value, the higher the dose reduction for a given therapeutic effect. Both CI and DRI, as well as concentration leading to 50% inhibition (IC_50_), were calculated using a median effect analysis by CompuSyn software (version 1.0.1, ComboSyn Inc., Paramus, NJ, USA). For the analysis, we chose the most effective sample in each assay (ABTS, DPPH, and DCFH-DA) and combined it with RA or LGlr (1:1 µg). We also tested a weight equivalent combination of RA and LGlr and their triple combination with the *Lycopus* leaf infusions. We did not test equimolar concentrations of RA and LGlr because we did not know the molar weights of lyophilisates. We used the same solutions in all experiments, so we chose the weight equivalent concentration.

## 5. Conclusions

Plants from the genus *Lycopus* may be considered as valuable antioxidant and antimicrobial agents. This study shows that the antioxidant activity of *Lycopus* infusions expressed as IC_50_ is lower than 10 µg/mL, which means a high antioxidant efficacy, especially for *L. exaltatus*, the antioxidant activity of which was studied for the first time. Furthermore, we detected a moderate synergistic effect between the most abundant compounds of infusion, namely rosmarinic acid and luteolin-7-*O*-glucuronide. A synergistic effect also occurs in the combination of infusion and rosmarinic acid or luteolin-7-*O*-glucuronide, respectively, and their triple combination. This interaction study, performed with *Lycopus* for the first time, reveals a high dose reduction index, which may lead to the possible use of lower doses of substances in combination in comparison to the doses of single substances. *Lycopus* infusions were also effective against diverse clinical *Staphylococcus aureus* strains with varying antimicrobial susceptibility, including methicillin-resistant ones. All these results suggest that the plants of the genus *Lycopus* will play an important role as promising antioxidants and antimicrobial agents in the future, especially in the treatment of local staphylococcal infections.

## Figures and Tables

**Table 1 molecules-25-01422-t001:** Content compounds of *Lycopus* leaf infusions, their corresponding retention times (T_R_), molecular ions [M − H]^−^, and MS^2^ fragments in LC-MS/MS analysis, in addition to quantitative abundance of polar phenolic compounds (µg/mL) in four *Lycopus* infusions.

	Compound	T_R_ (min)	[[M − H]^−^	MS^2^ (20 eV) (*m*/*z*)	Mass Concentration (µg/mL)* ± SD			
					LeuA	LeuB	LeuC	Lex
**1**	Citric acid	4.76	191.0191	111.0084 87.0087 85.037	ND	LOD	LOD	LOD
**2**	Danshensu (trihydroxyphenyl propanoic acid)	11.64	197.0449	179.0343 135.0443	LOD	LOD	LOD	LOD
**3**	Caffeic acid	21.59	179.0342	135.0445	15.7 ± 1.7	16.8 ± 0.2	16.9 ± 0.1	23.1 ± 0.1
**4**	Quercetin glucuronide	24.58	477.0656	301.0332	38.1 ± 1.4	21.5 ± 0.3	ND	84.3 ± 0.3
**5**	Quercetin glucoside	25.91	463.0857	301.0324	34.0 ± 7.7	LOD	LOD	LOD
**6**	Lithospermic acid	26.00	537.1034	359.0739 197.0456 179.0340 161.0235 135.0431	LOD	LOD	LOD	13.9 ± 0.1
**7**	Rosmarinic acid hexoside	28.42	521.1301	359.0767 179.0376 161.0223	LOD	17.4 ± 0.6	16.3 ± 0.5	LOD
**8**	**Luteolin-7-*O*-glucuronide**	28.57	461.0731	285.0393	**315.1 ± 3.6**	**279.6 ± 1.4**	**275.1 ± 0.9**	**449.8 ± 0.4**
**9**	Sagerinic acid	29.96	439.0318	359.0751 197.8040 161.0232	12.0 ± 2.4	15.4 ± 0.3	12.4 ± 0.2	11.1 ± 4.2
**10**	Sulphated rosmarinic acid	30.37	719.1620	539.1182 359.0774 161.0450	11.8 ± 0.6	18.7 ± 0.1	16.6 ± 0.4	13.0 ± 0.1
**11**	Apigenin-7-*O*-glucuronide	34.12	445.0753	269.0444	27.6 ± 0.1	23.0 ± 2.7	36.5 ± 1.2	27.2 ± 0.1
**12**	**Rosmarinic acid**	34.48	359.0751	197.0483 161.0234	**213.7 ± 0.3**	**237.5 ± 0.8**	**202.3 ± 0.1**	**216.2 ± 0.1**
**13**	Caffeic acid derivative	36.52	549.2332	359.0767 161.0295	19.6 ± 0.3	LOD	7.6 ± 0.2	168.0 ± 0.3

*Values (μg/mL infusion) are presented as means ± standard deviation (*n* = 3); external standards: luteolin-7-*O*-glucuronide (used for flavonoid determination), rosmarinic acid (used for phenolic acid determination). ND—not detected; LOD—limit of detection, T_R_ – retention time, LeuA—*L. europaeus* A; LeuB—*L. europaeus* B; LeuC—*L. europaeus* C; Lex—*L. exaltatus*;

**Table 2 molecules-25-01422-t002:** Activity of *Lycopus* infusions on the bacterial collection strains.

Strain	LeuA		LeuB		LeuC		Lex	
	MIC [mg/mL]	MBC [mg/mL]	MIC [mg/mL]	MBC [mg/mL]	MIC [mg/mL]	MBC [mg/mL]	MIC [mg/mL]	MBC [mg/mL]
*S. aureus* CCM 4223	2.5	2.5	2.5	2.5	2.5	2.5	5	5
*S. aureus* CCM 4750	2.5	2.5	2.5	2.5	2.5	2.5	2.5	2.5
*E. faecalis* CCM 4224	>40	>40	40	>40	40	>40	40	40
*P. aeruginosa* CCM 3955	20	40	40	>40	20	40	40	40
*E. coli* CCM 3954	40	40	40	>40	>40	>40	40	>40
*K. pneumoniae* CCM 4415	>40	>40	>40	>40	>40	>40	40	>40
*P. mirabilis* CCM 7188	5	5	2.5	2.5	2.5	5	5	10

Note: LeuA—*L. europaeus* A; LeuB—*L. europaeus* B; LeuC—*L. europaeus* C; Lex—*L. exaltatus*; MIC—minimal inhibitory concentration; MBC—minimal bactericidal concentration, CCM - Czech Collection of Microorganisms.

**Table 3 molecules-25-01422-t003:** Antimicrobial activity of *L. europaeus* B infusion against clinical *Staphylococcus aureus* strains, expressed as minimal inhibitory concentration and minimal bactericidal concentration versus number of inhibited and killed strains, respectively.

MIC [mg/mL]	Number of Strains (%)	MRSA (n)	MSSA (n)	MBC [mg/mL]	Number of Strains (%)	MRSA (n)	MSSA (n)
1.25	10 (36%)	8	2	1.25	9 (32%)	7	2
2.5	18 (64%)	6	12	2.5	18 (64%)	7	11
5	-	-	-	5	1 (4%)	-	1

MIC—minimal inhibitory concentration; MBC—minimal bactericidal concentration; n—number of strains; MRSA—methicillin-resistant *S. aureus*; MSSA—methicillin-susceptible *S. aureus.*

**Table 4 molecules-25-01422-t004:** The concentration of a sample necessary to decrease the oxidative stress by 50% (IC_50_) of plants infusions, rosmarinic acid, and luteolin-7-*O*-glucuronide by ABTS, DPPH, and DCFH-DA assay.

Sample	ABTS		DPPH		DCFH-DA	
	IC_50_ (μg/mL)	*r ^a^*	IC_50_ (μg/mL)	*r ^a^*	IC_50_ (μg/mL)	*r ^a^*
LeuA	9.35	0.99	8.42	0.99	2.64	0.98
LeuB	8.75	0.98	9.61	0.99	2.89	0.98
LeuC	9.61	0.96	9.02	0.98	2.92	0.99
Lex	10.41	0.97	8.30	0.98	2.17	0.97
LGlr	7.37	0.96	3.29	0.99	0.94	0.98
RA	2.20	0.97	1.81	0.99	0.63	0.99

*^a^ r*, linear correlation coefficient of the median-effect plot; LeuA—*L. europaeus* A; LeuB—*L. europaeus* B; LeuC—*L.europaeus* C; Lex—*L. exaltatus*; LGlr—luteolin-7-*O*-glucuronide; RA—rosmarinic acid; ABTS (2,2′-azino-bis(3-ethylbenzothiazoline-6-sulphonic acid), DPPH (2,2-diphenyl-1-picrylhydrazyl), DCFH-DA (dichlorodihydrofluorescein diacetate).

**Table 5 molecules-25-01422-t005:** Interaction analysis of *Lycopus spp.* infusions, luteolin-7-*O*-glucuronide, and rosmarinic acid.

Method	Sample Combination	IC_50_ (μg/mL) *^a^*	*r ^b^*	CI *^c^*	SDA *^d^*	Combined Effect	DRI *^e^*
ABTS	LeuB:RA	2.32	0.97	0.66	±0.01	Synergism	7.19:1.25
	LeuB:LGlr	6.64	0.98	0.83	±0.01	Moderate synergism	2.64:2.22
	LGlr:RA	2.74	0.98	0.81	±0.01	Moderate synergism	5.38:1.61
	LeuB:RA:LGlr	3.39	0.99	0.80	±0.01	Moderate synergism	7.74:1.95:6.52
DPPH	Lex:RA	2.75	0.96	0.93	±0.01	Nearly additive effect	6.03:1.31
	Lex:LGlr	2.96	0.97	0.63	±0.01	Synergism	5.60:2.22
	LGlr:RA	1.90	0.96	0.81	±0.03	Moderate synergism	3.46:1.90
	Lex:RA:LGlr	2.57	0.99	0.84	±0.01	Moderate synergism	9.70:2.12:3.85
DCFH-DA	Lex:RA	0.73	0.98	0.74	±0.01	Moderate synergism	6.00:1.74
	Lex:LGlr	0.98	0.98	0.75	±0.02	Moderate synergism	4.44:1.91
	LGlr:RA	0.54	0.96	0.72	±0.02	Moderate synergism	3.45:2.32
	Lex:RA:LGlr	0.58	0.99	0.61	±0.01	Synergism	11.16:3.23:4.80

*^a^* IC_50_ (μg/mL) of the equal mass concentration infusion combinations; ^b^ r—linear correlation coefficient of the median effect plot; ^c^ CI—combination index, a quantitative determination of drug interactions based on the mass action law, where CI < 1, = 1, and > 1 indicate synergism, additive effect, and antagonism, respectively (the combined effect is evaluated according to Chou [24]); ^d^ SDA—sequential deletion analysis, iterative sequential deletion of one dose (or concentration) of a drug at a time for repetitive CI calculations; ^e^ DRI—dose reduction index shows the extent that the dose of each drug in a synergistic combination can be reduced at a given effect level compared with the doses of each sample alone; LeuA—*L. europaeus* A; LeuB—*L. europaeus* B; LeuC—*L. europaeus* C; Lex—*L. exaltatus*; LGlr—luteolin-7-*O*-glucuronide; RA—rosmarinic acid.

**Table 6 molecules-25-01422-t006:** The harvest locations of plants.

	Latitude	Longitude	Elevation (m)
LeuA	50°52′30,8″	14°28′18,1″	380
LeuB	49°04′39.5″	14°20′45.9″	342
LeuC	46°07′51.42″	17°56′12.3″	174
Lex	47°54′27.00″	20°19′19.44″	324

LeuA—*L. europaeus* A; LeuB—*L. europaeus* B; LeuC—*L. europaeus* C; Lex—*L. exaltatus*.

**Table 7 molecules-25-01422-t007:** The tested bacterial collection strains.

Bacterial Species	The CCM / ATCC No of Strain	Note
*Staphylococcus aureus*	CCM 4223 / ATCC 29213	MSSA; ATM susceptibility QC strain
*Staphylococcus aureus*	CCM 4750 / ATCC 43300	MRSA; methicillin susceptibility reference strain
*Enterococcus faecalis*	CCM 4224 / ATCC 29212	ATM susceptibility QC strain
*Pseudomonas aeruginosa*	CCM 3955 / ATCC 27853	ATM susceptibility QC strain
*Escherichia coli*	CCM 3954 / ATCC 25922	ATM susceptibility QC strain
*Klebsiella pneumoniae*	CCM 4415 / ATCC 10031	ATM susceptibility QC strain
*Proteus mirabilis*	CCM 7188 / ATCC 29906	Assays of ATM preservative QC

Note: ATM—antimicrobial; QC—quality control; CCM—Czech Collection of Microorganisms; ATCC—American Type Culture Collection.

**Table 8 molecules-25-01422-t008:** The tested clinical *Staphylococcus aureus* strains.

Strain	Origin	Antimicrobial Susceptibility							*mecA* Gene	Spa-Type
		OXA	ERY	CLI	TET	COT	CIP	MUP		
1	Skin, soft tissue, or wound infection	S	R	R	S	S	S	S	−	t571
2		S	R	R	S	S	S	S	−	t024
3		S	R	R	S	S	S	S	−	t084
4		S	R	R	S	S	R	R	−	t2716
5		S	R	R	R	S	S	S	−	t160
6		S	S	S	S	S	S	S	−	t1309
7		S	R	R	R	S	R	R	−	t2448
8		S	S	S	S	S	S	S	−	t1491
9		S	S	S	S	S	S	S	−	t065
10		S	R	R	S	S	R	S	−	t4559
11		R	R	R	R	S	R	S	+	t032
12		R	S	S	S	S	R	S	+	t032
13		R	R	R	S	S	R	S	+	t189
14		R	R	R	S	S	R	S	+	t003
15		R	R	S	S	S	R	S	+	t008
16		R	R	R	S	S	R	S	+	t718
17		R	R	R	S	S	R	S	+	t003
18		R	R	R	S	S	R	S	+	t1148
19		R	R	R	S	S	R	S	+	*
20	CVC	R	S	S	S	S	R	S	+	t032
21	HC	S	R	S	R	S	S	S	−	t024
22		S	R	S	S	S	S	S	−	t036
23		S	R	R	S	S	S	S	−	t1451
24		S	R	R	S	S	R	S	−	t3723
25		R	R	R	I	S	R	S	+	t003
26		R	R	R	S	S	R	S	+	t014
27		R	S	S	S	S	R	S	+	t032
28		R	R	R	S	S	R	S	+	t4559

Note: CVC—central venous catheter; HC—hemoculture; OXA—oxacillin; ERY—erythromycin; CLI—clindamycin; TET—tetracycline; COT—cotrimoxazole; CIP—ciprofloxacin; MUP—mupirocin; *mecA* gene coding for PBP2a protein, responsible for methicillin resistance in staphylococci; +—positive; * new spa type, uploaded to the international database and waiting for t-number assignment; S—susceptible, standard dosing regimen; R—resistant; I—susceptible, increased exposure; the corresponding MIC values are included in Table 9.

**Table 9 molecules-25-01422-t009:** The MIC values corresponding to the susceptibility categories (S and R) of the tested antibiotics, based on the EUCAST (The European Committee on Antimicrobial Susceptibility Testing) clinical break-points [52].

Antibiotic	Susceptible [mg/L]	Resistant [mg/L]
Oxacillin	<2	>2
Erythromycin	≤1	>2
Clindamycin	≤0.25	>0.5
Tetracycline	≤1	>2
Cotrimoxazole	≤2	>4
Ciprofloxacin	≤0.001	>1
Mupirocin	≤1	NA

NA—not applicable; only screening.

**Table 10 molecules-25-01422-t010:** The scales of combined effects using the CI grading.

Range of Combination Index	Description
<0.1	Very strong synergism
0.1–0.3	Strong synergism
0.3–0.7	Synergism
0.7–0.85	Moderate synergism
0.85–0.90	Slight synergism
0.90–1.10	Nearly additive
1.10–1.20	Slight antagonism
1.20–1.45	Moderate antagonism
1.45–3.3	Antagonism
3.3–10	Strong antagonism
>10	Very strong antagonism

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
