# Peer review of "Antimicrobial and Antioxidant Properties of Four Lycopus Taxa and an Interaction Study of Their Major Compounds"

_molecules, 2020, doi:10.3390/molecules25061422_

Round 1
Reviewer 1 Report
Dear authors,
The work entitled "Antimicrobial and Antioxidant Properties of Four Lycopus taxa and an Interaction Study of Their Major Compounds" contributes to the chemical prospection of medicinal plants and to the understanding of their pharmacological properties. However, here are some suggestions:
I believe that the observed antimicrobial activities (MIC - minimal inhibitory concentration, and MBC - minimal bactericidal concentration) were only obtained if using high concentrations of the test samples.
In the discussion on antimicrobial activity, the role of secondary metabolites of Lycopus infusions and how this can affect activities on Gram positive and Gram negative microorganisms is not highlighted.
According to the values of MIC - minimal inhibitory concentration, and MBC - minimal bactericidal concentration, the manuscript needs to show, according to the literature, which potency an extract must present in order to become a valuable antimicrobial agent.
Author Response
Dear reviewer, thank you for the time you paid to review our manuscript, as well as for all of your suggestions on how to improve its text.
Here below please see our reply:
I believe that the observed antimicrobial activities (MIC - minimal inhibitory concentration, and MBC - minimal bactericidal concentration) were only obtained if using high concentrations of the test samples.
Antimicrobial activities were obtained using concentrations given in the tables and the text – i.e. the bacterial inocula of S. aureus strains used in the tests were inhibited in their growth at concentrations of the Lycopus infusions ranging from 1250 to 2500 mg/ml - these were the MICs. The used S. aureus bacteria were killed at the concentrations from 1250 to 5000 mg/ml - these were the MBCs (the exact definition of the MBC: the lowest concentration of the antimicrobial agent, at which 99.9 % of the used bacterial inoculum is killed).
In the discussion on antimicrobial activity, the role of secondary metabolites of Lycopus infusions and how this can affect activities on Gram positive and Gram negative microorganisms is not highlighted.
We have not tested the interaction of the individual secondary metabolites contained in the tested Lycopus infusions with the cell walls of the microbial genera mentioned in the text of the manuscript. In the discussion, we expressed only our hypothesis. Therefore, we included an additional explanation of the text:
„ Differences in the susceptibility of various bacterial species were probably based on the well-known differences in their cell walls, such as different permeability, due to the different structure of Gram-positive and Gram-negative bacterial cell wall, charge of the lipopolysaccharide molecules in the Gram-negative cell walls (e.g. Proteus spp. versus the majority of the other Gram-negative bacteria), or an insufficient redox potential on the cytoplasmic membrane in Enterococcus spp. The reason for the resistance of the tested Candida albicans strain might be of similar origin (impermeable cell wall). However, we have not evaluated the interaction of the individual secondary metabolites contained in the Lycopus infusions with the cell walls of the mentioned microbes. “
According to the values of MIC - minimal inhibitory concentration, and MBC - minimal bactericidal concentration, the manuscript needs to show, according to the literature, which potency an extract must present in order to become a valuable antimicrobial agent.
The acceptable potency of antimicrobial agent depends on its toxicity toward humans and on the intended application form. The first property will be tested in the following studies. Based on the obtained MICs/MBCs of the tested Lycopus infusions, we consider the topical application as the appropriate form - pls see the text: „Taking into account the MIC/MBC values of tested Lycopus infusions for the bacterial strains used in the study, the infusions might be effective especially in the treatment of local infections. As it is known, the topical remedies application enables an immediate contact of active compounds, at rather high concentrations, with the microorganisms present in the infectious focus. “
Reviewer 2 Report
This paper explores the bioactivites of extracts of four plant species. Using LC-MS/MS the major components were identified and quantified with two major components known. The four extracts and pure samples of these known compounds (RA and LGlr) were tested for antibacterial and antioxidant activity. Subsequently the two most 'active' extracts then underwent interaction analysis.
The major issue is over analysis.
It is often not clear what is new in this study. Of course one of the extracts in new but what has been done already with the others. Have MS-MS studies already been done on these compounds? if so do the authors need to go into detail. Or are there some that have not been reported. Or is the MS-MS incomplete? Table 1- What was the LOD? there is no LOQ in the table but there is a ND. Perhaps the important results are lost in the detail. Does another table 'summarising' table 1 need to go in to highlight the key results, or does table 1 one need to be pruned (eg elimination rows 1 and 2, MS-MS fragments...molec ion)
The antimicrobial activity of these compounds is poor. There is no positive or negative control included in Table 2. The 'most active' compound has an MIC of 2500 ug/ml!!. how does this compare to a gold standard antibiotic? the discussion on which were more susceptible is pretty much irrelevant as they are not really active anyway.
i do not understand what table 3 is trying to explain.. at all..something about they are active against drug resistant antibiotics. again the conc are really high which again undermines the use of these infusions in the clinic. How does the conc needed for bioactivity relate to the conc of components in the extract?
antioxidant activity- some nice results but again over analysed. There are subtle differences in activity but no orders of magnitude like. I am not sure how accurate these tests are but to report to 2 decimal places with no error! A +- number probably means more to a reader than a r value
Probably the most interesting result was the interaction analysis. But have results been cherry-picked? They left out the results of the interaction analysis between different infusions (whatever that means) due to only observing additive effects.
Questions on methodology
What was the impact of drying the aerial parts in the dark? How long did this take? Rosmarinic acid is susceptible to decomposition depending on drying conditions. It would be interesting to see what level of RA is present in fresh leaves. https://doi.org/10.1080/0972060X.2017.1413957
Overall grammar and spelling need a lot or work. Far too many to list individually- here is some from the intro
- remove first two sentences in abstract
- poor use of commas in intro
- line 49 fullstop after effect and delete the next two words
- line 51 effectivity to efficacy?
- line 52 Due to the decrease or elimination... does not make sense
- line 57 herbal juice yield a black dye? Does this mean juicing/mincing the leaves gives a black
- line 58 delete ones
- line 63 is the comma after essential oils needed? brackets around the mostly diterpenes and triterpenes
- line 74 replace conventionally suggested with traditionally used
- line 76 what does it do to body temp
- lines 85 and 92 mean the same thing
- line 85 remove yet
- line 88 delete the in front of human
- line 107- what literature?
- throughout leaves infusion with leaf infusion?
Author Response
Dear reviewer, thank you for the time you paid to review our manuscript, as well as for all of your suggestions on how to improve its text.
Here below please see our reply:
The major issue is over analysis.
It is often not clear what is new in this study. Of course one of the extracts in new but what has been done already with the others. Have MS-MS studies already been done on these compounds? if so do the authors need to go into detail. Or are there some that have not been reported. Or is the MS-MS incomplete?
We are not sure if we exactly understand, but in the case of herbal extract (even known plants) is the MS/MS always most valuable and helpful detection. There is not so much available literature on polar phenolics in Lycopus leaves, but we published also an analysis in the past (see Ref 13). The novelty is especially about L. exaltatus (as you mentioned). Anyway, the samples of L. europaeus origin from nature (not planted) and it is visible that they also differ in presented phenolics and their content, also in comparison to our previous studies on secondary metabolites determination. MS/MS is complete, all details are explained in discussion.
Table 1- What was the LOD?
Both LOQ and LOD values are mentioned in the text below. Please see lines 147-149.
there is no LOQ in the table but there is a ND.
Corrected, LOQ deleted, ND inserted
Perhaps the important results are lost in the detail. Does another table 'summarising' table 1 need to go in to highlight the key results, or does table 1 one need to be pruned (eg elimination rows 1 and 2, MS-MS fragments...molec ion)
We highlighted the important results in Table 1. To express the activity we count especially with two major compounds. The all-important results are displayed in Table 1.
The antimicrobial activity of these compounds is poor.
Please see the explanation below
There is no positive or negative control included in Table 2.
Performing the antimicrobial activity tests, which yielded the results in table 2, we used the bacterial growth medium free of antimicrobial agents as a positive control, and the medium with the tested agents free of bacterial inoculum as a negative control - as it is stated in the manuscript. To make this fact more evident, we included expressions „negative controls“ and „positive controls“ to the text:
„The standardized suspensions of microorganisms were added to each well in 10 µL aliquots (except for the sterility control wells – negative controls) and incubated for 24 hours at 35 °C. Wells with microorganisms in the medium free of the tested agent were used as growth control (positive controls).“
The 'most active' compound has an MIC of 2500 ug/ml!!. how does this compare to a gold standard antibiotic?
The tested Lycopus water infusions had bactericidal activity against collection strains of staphylococci at 2500 mg/ml, (i.e. 2.5 mg/ml), and in the case of clinical S. aureus strains the MBCs were in the range from 1250 to 5000 mg/ml, (i.e. from 1.25 mg/ml to 5 mg/ml). This concentration, if we suppose the absence of toxic or irritant effects during topical application (the adverse effects should be excluded during further studies), could be effective at least in a form of topical antistaphylococcal agent, active even against methicillin-resistant and polyresistant strains (for comparison pls see the content of clindamycin in a topical agent Dalacin, which contains 10 mg of antibiotics per 1 ml of emulsion). This fact is stated in our manuscript as well: „Taking into account the MIC/MBC values of tested Lycopus infusions for the bacterial strains used in the study, the infusions might be effective especially in the treatment of local infections. As it is known, the topical remedies application enables an immediate contact of active compounds, at rather high concentrations, with the microorganisms present in the infectious focus.“
the discussion on which were more susceptible is pretty much irrelevant as they are not really active anyway.
The MBCs against the other tested collection strains were really rather high, however, in the case of P.mirabilis they still could be taken in account (2500 mg/ml – i.e. 2.5 mg/ml in the case of the infusion LeuB, and 5000 mg/ml – i.e. 5 mg/ml in the case of the infusion LeuA and LeuC). Therefore, we do not consider the discussion on it to be irrelevant.
i do not understand what table 3 is trying to explain.. at all..
Table 3 is trying to explain, that the infusion LeuB (the most active infusion from the 4 tested ones) has antibacterial activity also against clinical S. aureus strains of wide clonal diversity and various antimicrobial resistance pattern. To be clearer, we more specified table 3.
Table 3 Antimicrobial activity of L. europaeus B infusion against clinical Staphylococcus aureus strains, expressed as minimal inhibitory concentration / minimal bactericidal concentration versus the number of inhibited/killed strains
|
MIC [µg/mL] |
Number of strains (%) |
MRSA (n) |
MSSA (n) |
MBC [µg/mL] |
Number of strains (%) |
MRSA (n) |
MSSA (n) |
|
1250 |
10 (36 %) |
8 |
2 |
1250 |
9 (32 %) |
7 |
2 |
|
2500 |
18 (64 %) |
6 |
12 |
2500 |
18 (64 %) |
7 |
11 |
|
5000 |
- |
- |
- |
5000 |
1 (4 %) |
- |
1 |
MIC – minimal inhibitory concentration; MBC – minimal bactericidal concentration; n – number of strains; MRSA – methicillin-resistant S. aureus; MSSA – methicillin-susceptible S. aureus
something about they are active against drug resistant antibiotics.
In the text of the manuscript, there is not any reference on „activity against drug-resistant antibiotics“.
again the conc are really high which again undermines the use of these infusions in the clinic.
The explanation about the „really high concentrations“ and the potential usage of Lycopus infusion in clinical practice is given above.
How does the conc needed for bioactivity relate to the conc of components in the extract?
Well, from the pharmacognostic point of view it is hard to define. The extract is a mixture of compounds, not only RA will act, that we know... As we did the antimicrobial testing on RA in the past, we added this information into Discussion (the conc. of extract should be about 3-4x higher).
antioxidant activity- some nice results but again over analysed. There are subtle differences in activity but no orders of magnitude like. I am not sure how accurate these tests are but to report to 2 decimal places with no error! A +- number probably means more to a reader than a r value
In synergy studies, the choice of drug–effect relationship is the basis for 50% level effect calculation, which is later used for interaction analysis. As we chose to use the median effect equation, derived by Chou for synergy quantification, we used their CompuSyn software too, where the drug-effect function, combination index, and statistics were calculated. Therefore statistical analysis was performed by the CompuSyn software within its own mathematical approach. For the IC50 was done using r value: the conformity parameter for the goodness of fit to the median-effect principle (MEP) of the mass-action law. It is the linear correlation coefficient of the median effect plot, where r 1 indicates perfect conformity (Chou 2006).
Chou TC (2006) Theoretical Basis, Experimental Design, and Computerized Simulation of Synergism and antagonism in Drug Combination Studies. Pharmacol Rev 58:621–681, 2006.
Probably the most interesting result was the interaction analysis. But have results been cherry-picked? They left out the results of the interaction analysis between different infusions (whatever that means) due to only observing additive effects.
We combined infusions LeuA, LeuB, LeuC and Lex (1:1) and in all combinations consisting of two infusions we observed just additive effect. It means, that there is no interesting interaction between these infusions, so we did not present these data, as is also written in manuscript in the lines 293-294: „We provided also the interaction analysis between different infusions but was observed additive effect only, therefore we did not present here these results.“ We considered this data “not interesting and innovative”, so we did not present them in the manuscript and we focused on the interaction study between rosmarinic acid and luteolin-7-O-glucuronide, which are the most abundant compound in Lycopus infusion.
Questions on methodology
What was the impact of drying the aerial parts in the dark? How long did this take? Rosmarinic acid is susceptible to decomposition depending on drying conditions. It would be interesting to see what level of RA is present in fresh leaves. https://doi.org/10.1080/0972060X.2017.1413957
The drying was made at room temperature as prescribed in the European Pharmacopoeia 10th edition. The aim of this study was not to compare drying methods but compare plant material dried at the same conditions. Drying at room temperature without sunlight is the most common method of drying. The fresh material contains ballast substances and water, which may influence used weights and counting. It could be interesting to compare RA of fresh and dried, but maybe in another study.
Overall grammar and spelling need a lot or work. Far too many to list individually- here is some from the intro
remove first two sentences in abstract
Removed
line 107- what literature?
added
line 49 fullstop after effect and delete the next two words
Corrected
line 51 effectivity to efficacy?
Done
line 52 Due to the decrease or elimination... does not make sense
Corrected
line 57 herbal juice yield a black dye? Does this mean juicing/mincing the leaves gives a black
Exactly
line 58 delete ones
Done
line 63 is the comma after essential oils needed? brackets around the mostly diterpenes and triterpenes
A comma is needed I think, brackets inserted
line 74 replace conventionally suggested with traditionally used
Done
line 76 what does it do to body temp
It decreases body temperature, which is increased due to hyperthyreosis. We cleared it in text.
lines 85 and 92 mean the same thing
One of the sentences is deleted
line 85 remove yet
Removed
line 88 delete the in front of human
Deleted
throughout leaves infusion with leaf infusion?
Corrected
Reviewer 3 Report
The manuscript entitled "Antimicrobial and Antioxidant Properties of Four Lycopus Taxa and an Interaction Study of Their Major Compounds" presents a topic of current interest. I suggest the authors to improve the presentation quality to make the reading more stimulating and to make the content of the study more appreciated, which is a bit lost.
There are grammatical mistakes along the manuscript, for examples : lines 65-66
line 103: TR, R must be subscript
line 148: “μg per mL of lyophilised infusion”….if lyophilised maybe gram or mg….not ml. In table 1 the authors affirm μg/mL infusion.
line 276: heterogenecity could be replaced by heterogeneity
Check the character of equation 1 and correct “Eqation”, the same for Equation 2
Author Response
Dear reviewer, thank you for the time you paid to review our manuscript, as well as for all of your suggestions on how to improve its text.
Please see our reply below:
I suggest the authors to improve the presentation quality to make the reading more stimulating and to make the content of the study more appreciated, which is a bit lost.
Thank you very much for your suggestion. We tried to make the reading more comprehensive by dividing the text in the Results section and the Discussion section. We hope that it improved the presentation quality.
line 148: “μg per mL of lyophilised infusion”….if lyophilised maybe gram or mg….not ml. In table 1 the authors affirm μg/mL infusion
corrected
line 103: TR, R must be subscript
Done
line 276: heterogenecity could be replaced by heterogeneity
Done
Check the character of equation 1 and correct “Eqation”, the same for Equation 2
Corrected
Round 2
Reviewer 1 Report
Dear authors,
The manuscript "Antimicrobial and Antioxidant Properties of Four Lycopus taxa and an Interaction Study of Their Major Compounds" presents a good chemical study, with identification of many constituents of the plant. In other words, the phytochemical part is adequate and in fact contributes to the study of Lycopus.
However, the antimicrobial activity of infusions is only obtained in high concentrations. Therefore, in view of the microorganisms tested, the samples did not reveal antimicrobial potential.
In addition, there is no in-depth description in the discussion relating the main secondary metabolites identified and the antimicrobial activities observed. This is essential for this type of scientific journal.
Finally, the study also lacks a description of the toxicological potential of the samples.
Author Response
Dear reviewer,
Thank you for your opinion of our work.
We understand you...it could be amazing to find a new strong antimicrobial drug, but let´s look at it from the pharmacognostic point of view. We do not develop antibiotics (not that we don't want to), this is just basic research on herbal infusion´s antimicrobial action, as it is explained in the manuscript. Yes, if we would test pure secondary metabolites (rosmarinic acid, or luteolin derivatives, or other flavonoids or phenolics) we can be in around 10times better activity, but we tested a water extract, a mixture of mostly polar phenolics, which have high antioxidant potential and can contribute on wounds healing. The antimicrobial testing of pure substances and their mutual activity in equimolar mixtures will be a target for future study.
A common tea (= infusion) of one of the Lamiaceae plants. We do not expect any toxicity, only in extra high doses or maybe in case of really long term oral use. According to a German monograph on Lycopus, for preparing water infusion is the maximal dose of 2 grams of plant per day. According to our yield of extraction (around 20%), we can predict, that the infusion will have around 400 mg of active compounds after lyophilization. The highest concentration we tested was 40 mg/ml, so it is 10 times lower than the maximal daily dose (still not toxic) for oral consumption. Anyway, in this case, it is irrelevant. We write in the text, that infusions are for local use, as we worked with clinical S. aureus strains from SKIN lesions and wounds, where all samples were active, please see table 3. In table 2 is the activity of infusions on collection strains. Although we see lower activity against G-, we cannot skip this important information.
Reviewer 2 Report
The main (minor) issue remains with controls. Why not include a column in table 2 for the control as a comparison. Looking at table 2 I cannot tell if these are exciting results as I dont know how they compared to a reasonable antibiotic (without trolling through the text).
To improve readability why not convert these to mg/ml- it would cut out a lot of zeros......
Author Response
Dear Reviewer,
thank you for the time you paid to review our manuscript, as well as for all of your suggestions on how to improve its text. The information about the changes we made and the explanations of our intention follows:
The main (minor) issue remains with controls. Why not include a column in table 2 for the control as a comparison. Looking at table 2 I cannot tell if these are exciting results as I dont know how they compared to a reasonable antibiotic (without trolling through the text).
Antibiotic control was not included in Table 2, as the strains are well-characterized collection strains. They were used for the first screening; this information was added to the text of manuscript: „For the first screening, 7 well-characterized collection strains were used (for details see the “Material and Methods”).“, and the collection strains were better characterized in the Material and Methods.
Based on the results with collection strains, clinical S. aureus strains were used for further testing. The table with their characteristics in the Material and Methods was supplemented wit clinical break-point levels of the Susceptible and the Resistant categories.
To improve readability why not convert these to mg/ml- it would cut out a lot of zeros......
Thank you for the suggestion, the numbers were converted to mg/ml.